# Back Razor: Memory-Efficient Transfer Learning by Self-Sparsified Backpropagation

**Ziyu Jiang**[†][*]**, Xuxi Chen**[‡][*]**, Xueqin Huang**[†]**, Xianzhi Du**[§]**, Denny Zhou**[§]**, Zhangyang Wang**[‡]
[†]Texas A&M University    [‡]University of Texas at Austin    [§]Google
{jiangziyu,xueq13}@tamu.edu,{xxchen,atlaswang}@utexas.edu
{xianzhi,dennyzhou}@google.com

## Abstract

Transfer learning from the model trained on large datasets to customized downstream tasks has been widely used as the pre-trained model can greatly boost the generalizability. However, the increasing sizes of pre-trained models also lead to a prohibitively large memory footprints for downstream transferring, making them unaffordable for personal devices. Previous work recognizes the bottleneck of the footprint to be the activation, and hence proposes various solutions such as injecting specific lite modules. In this work, we present a novel memory-efficient transfer framework called **Back Razor**, that can be plug-and-play applied to any pre-trained network without changing its architecture. The key idea of Back Razor is *asymmetric sparsifying*: pruning the activation stored for back-propagation, while keeping the forward activation dense. It is based on the observation that the stored activation, that dominates the memory footprint, is only needed for back-propagation. Such asymmetric pruning avoids affecting the precision of forward computation, thus making more aggressive pruning possible. Furthermore, we conduct the theoretical analysis for the convergence rate of Back Razor, showing that under mild conditions, our method retains the similar convergence rate as vanilla SGD. Extensive transfer learning experiments on both Convolutional Neural Networks and Vision Transformers with classification, dense prediction, and language modeling tasks show that Back Razor could yield up to **97% sparsity**, saving **9.2x memory** usage, without losing accuracy. The code is available at: https://github.com/VITA-Group/BackRazor_Neurips22.

## 1 Introduction

Edge devices equipped with deep learning applications are becoming ubiquitous. However, only deploying pre-trained models is not sufficient: these edge devices may keep collecting novel data that general pre-trained models have never seen before, and fine-tuning on them would notably improve the model's performance. A typical approach is to upload these newly collected data to cloud servers for fine-tuning, and download the trained model from the cloud to local devices. However, such a way has potential privacy issues due to data sensitivity and also imposes undesirable pressures on the Internet bandwidth, brought by transmitting both the collected data and trained models. These challenges motivate researchers to explore *on-device learning* methods, i.e., how to fine-tune pre-trained models on memory-limited devices.

One of the main challenges of fine-tuning on edge devices is the memory constraints. Figure 1 exemplifies the memory usage of training some popular networks: ResNet50 with a small batch size of 16 can easily exceed the memory limitation of onboard devices. The newly emerged Vision

---

[*]Equal contribution

Transformer (ViT) would incur even more memory consumption. Recent studies show that the activations stored for back-propagation take up a large proportion of memory cost during training. Therefore, popular model compression techniques such as weight pruning and quantization are not highly efficient in reducing training memory footprints. Parameter efficient training method like Bitfit also fails to compress the activation to memory limitation (see Figure 1). Gradient checkpointing can reduce the training memory by storing only a subset of activations, but requires more FLOPs at back-propagation due to the re-computation of discarded activations. Using a smaller batch size or half-precision for training can also lower the memory cost, but they incur a decrease in the model's accuracy and training speed. [1] successfully avoided saving activation via freezing the weights for convolutional neural networks and only learning a newly added the lite residual module, thus no need to store the intermediate activations. However, it comes at a certain price of the transfer performance.

A new stream of works studied how to directly compress the activations. They mainly leverage the quantization technique that discretizes the activation tensors so these tensors can be represented by efficient data formats and require less memory [2, 3, 4]. However, the activation pruning techniques have been less explored. [5] applied random activation masks on each layer in both forward and backward passes to reduce latency. [6] integrated weight pruning and activation pruning by learning importance scores on neuron's outputs. Moreover, they adopt the symmetric scheme that prunes the activation at both the forward and backward processes. Such a pruning scheme undermines the precision of models' outputs, leading to significantly worse performance when the pruning ratio is high.

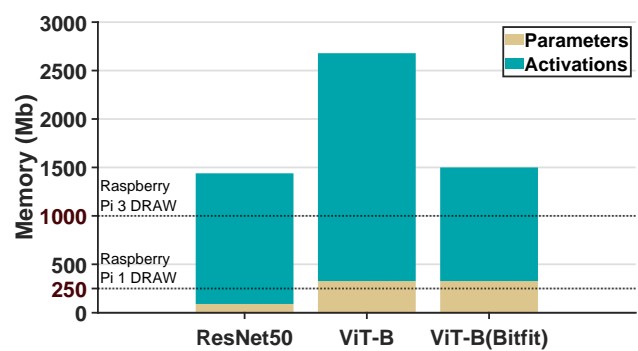

Figure 1: Training memory usage and on-device memory limitation illustration. The training memory is calculated under a batch size of 16. ViT-B (Bitfit) denotes the ViT-B model tuned with Bitfit.

In this work, we present a general activation compression method called **Back Razor**. Inspired by the observations that the activations are only stored for back-propagation, we adopt the asymmetric pruning scheme that only prunes the activation during the backward process. Specifically, after activation is used by a subsequent layer, our method replaces it with a pruned activation which carries a lower memory cost. During the backward phase, the gradients are calculated with the approximated activation, but only with a small error between those with the *exact* activation.

Our contributions are summarized as following:

- We propose a memory-efficient transfer learning method called Back Razor that prunes the activation for back-propagation. Our proposed method can be easily applied to various backbones, including convolutional neural networks (CNNs) and vision transformers (ViTs).

- We provide a theoretical analysis of our self-sparsified back-propagation method for CNNs. Specifically, we discover that under mild assumptions, our method can share a similar convergence rate as vanilla SGD.

- We extensively verify the proposed Back Razor on both CNNs and ViTs for multiple datasets and show that the proposed method can achieve up to 97% sparsity with saving $9.2\times$ memory saving without losing accuracy. By further verifying on GPU, the proposed method achieves $2.7\times$ on-device memory efficiency for ViTs.

## 2 Related Works

### 2.1 Pruning and Sparse Training

Pruning can be operated at different levels, *e.g.*, pruning on weight, weight gradients, activations, and activation gradients. Pruning on weights are most widely studied since it can accelerate the inference

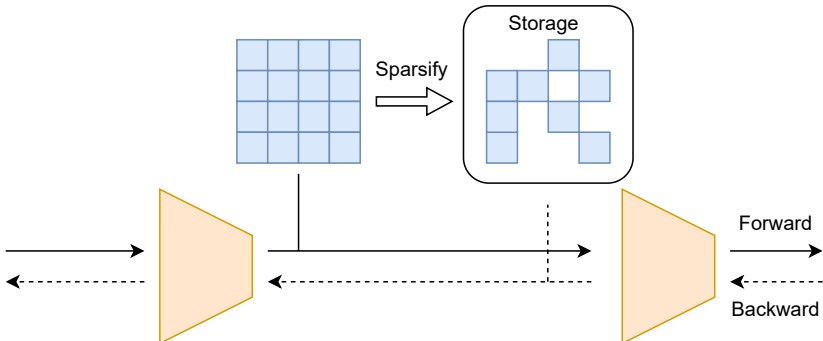

Figure 2: Pipeline of the proposed Back Razor.

phase [7, 8, 9]; pruning on weight gradients can help reduce communication cost especially in distributed training [10, 11]; pruning on activation gradients [12, 13, 14, 15] or activation itself have been less addressed since they are mostly dynamic during training. [5] randomly generated static sparse masks and applied them on each layer in both forward and backward passes. [6] proposed integral pruning that prunes both activation and weights by consecutively learning and applying two masks on weights and activations. However these works are adopting the symmetric pruning scheme, *i.e.*, pruning the activation at both the forward and the backward process. Pruning the activations at each layer brings errors, and they will accumulate as the forwarding proceeds. Backdrop [16] improves the generalizability of large batch gradient descent via randomly masking parts of the backward gradient propagation. SWAT [17] empirically explores sparsifying both weights and activations for training CNNs from scratch, and the authors also discovered that pruning activations only for backprogration can effectively preserve accuracy better. While our idea shares similarity to SWAT, Back Razor is dedicated to transfer learning and is applicable to general backbones (both CNNs and ViTs), and we also supply the theoretical analysis for such self-sparsified backpropogation for the first time.

## 2.2 Memory Efficient Transfer Learning

Transfer Learning from a model pre-trained on large datasets to small datasets has been widely used for many applications [18, 19, 20, 21]. In contrast to the pre-training that is always conducted on large clusters, transfer learning may happen on personal computing devices with limited memory constrains. To fit the large model to small devices, a common choice is to fine-tune the last several layers of the model [22, 18, 23, 24]. However, it can be a sub-optimal choice, especially for the downstream tasks with large distribution differences. In contrast, fine-tuning the whole can yield better accuracy in most cases [1, 25]. However, it can easily lead to memory outages. On the other hand, recent parameter efficient transfer learning algorithms [26, 27, 28, 29] only require tuning less than $1\%$ of the parameters. But this parameter efficiency only brings a mild memory efficiency. The most relevant prior work is TinyTL [1], it recognizes the main bottleneck of memory is in activation and introduces a new trainable module that only requires down-sampled activation. However, it still focuses on exact gradient calculation while we employ approximate gradient calculation, which enables compressing activation without changing the model architecture. A few other works have also explored the approximate activation techniques. Most of them focus on activation quantization [2, 3, 30, 31]. Recently, activation compression is extended to ViTs [32] and Graph Neural Networks [33].

## 3 Method

### 3.1 Activation is the bottleneck of memory

The forward of neural networks can be express as following

$$f(\boldsymbol{x}; \boldsymbol{\theta}) = f_l \left( f_{l-1} \left( \ldots f_0 \left( \boldsymbol{x}; \boldsymbol{\theta}_0 \right) \ldots ; \boldsymbol{\theta}_{l-1} \right) ; \boldsymbol{\theta}_l \right)$$
$$L = \mathcal{L}(f(\boldsymbol{x}; \boldsymbol{\theta}), y) \tag{1}$$

where $\boldsymbol{x}$ and $y$ is the input and label, respectively. $L$ is the loss between prediction $f(\boldsymbol{x}; \boldsymbol{\theta})$ and label $y$ calculated by loss function $\mathcal{L}$. $f_i$ and $\boldsymbol{\theta}_i$ denote the function and parameters of $i$ th layer, respectively. The network is composed of $l$ layers. We further denote the output of $i$ th layer (a.k.a activation) as $\boldsymbol{z}_i = f_i(\boldsymbol{z}_{i-1}, \boldsymbol{\theta}_i)$ and $\boldsymbol{z}_1 = f_1(\boldsymbol{x}; \boldsymbol{\theta}_1)$. When conducting back-propagation, $\frac{\partial L}{\partial \boldsymbol{z}_i}$ and $\frac{\partial L}{\partial \boldsymbol{\theta}_i}$ are required to be calculated for each activation.

Taking linear layer with $\boldsymbol{z}_i = \boldsymbol{z}_{i-1} W_i + b_i$ ($\boldsymbol{\theta}_i = [W_i, b_i]$) as an example, the gradients required to be calculated are [1]:

$$\frac{\partial L}{\partial \boldsymbol{z}_i} = \frac{\partial L}{\partial \boldsymbol{z}_{i+1}} \frac{\partial \boldsymbol{z}_{i+1}}{\partial \boldsymbol{z}_i} = \frac{\partial L}{\partial \boldsymbol{z}_{i+1}} W_i^\top, \quad \frac{\partial L}{\partial W_i} = \boldsymbol{z}_i^\top \frac{\partial L}{\partial \boldsymbol{z}_{i+1}}, \quad \frac{\partial L}{\partial b_i} = \frac{\partial L}{\partial \boldsymbol{z}_{i+1}} \tag{2}$$

As shown in Equation 2, the activation $\boldsymbol{z}_i$ is employed for computing one of the gradients. This creates the need of saving the activation of the forward process. Otherwise, extra FLOPs are required for re-computing the activation. The size of activation is always larger than the weight as weight is always shared across different patches or tokens. Also, the size of activation can increase linearly with the batch size. Moreover, in the transformer, the length of input tokens can also lead to a quadratic increase of the memory.

**Discussion about freezing weight.** TinyTL [1] pointed out that we can avoid saving activation via freezing the weight $W_i$ as other terms do not require the participation of $\boldsymbol{z}_i$. This is true for most architectures of convolutional neural networks. However, the emergence of ViT breaks this assumption. Many operations (e.g. self-attention, Softmax, GeLU [34]) in transformer would involve the activation for computing $\frac{\partial L}{\partial \boldsymbol{z}_i}$, which cannot be avoided if previous layers requires updating. Therefore we adopt the more general method: directly sparsifying the activation for backpropagation.

## 3.2 Back-propagation activation self-sparsification

In contrast to the activation sparsification [5, 6] that prunes the activation of both forward and backward, our proposed **Back Razor** leverages the **back-propagation** activation sparsification: it only prunes the activation that is used for backward while keeping the forward activation dense as shown in Figure 2. The normal backward process can be formally defined as

$$\frac{\partial L}{\partial \boldsymbol{z}_i} = \frac{\partial L}{\partial \boldsymbol{z}_{i+1}} \frac{\partial \boldsymbol{z}_{i+1}}{\partial \boldsymbol{z}_i} = \frac{\partial L}{\partial \boldsymbol{z}_{i+1}} h_i^{(z)}(\boldsymbol{z}_i, \boldsymbol{\theta}_i), \quad \frac{\partial L}{\partial \boldsymbol{\theta}_i} = \frac{\partial L}{\partial \boldsymbol{z}_{i+1}} \frac{\partial \boldsymbol{z}_{i+1}}{\partial \boldsymbol{\theta}_i} = \frac{\partial L}{\partial \boldsymbol{z}_{i+1}} h_i^{(\theta)}(\boldsymbol{z}_i, \boldsymbol{\theta}_i) \tag{3}$$

where $h_i^{(z)}(\boldsymbol{z}_i, \boldsymbol{\theta}_i) := \frac{\partial \boldsymbol{z}_{i+1}}{\partial \boldsymbol{z}_i}$ and $h_i^{(\theta)}(\boldsymbol{z}_i, \boldsymbol{\theta}_i) := \frac{\partial \boldsymbol{z}_{i+1}}{\partial \boldsymbol{\theta}_i}$, representing the derivatives of $\boldsymbol{z}_{i+1}$ with respect to $\boldsymbol{z}_i$ and $\boldsymbol{\theta}_i$, respectively. For Back Razor, the backward process can be expressed as

$$\frac{\partial L}{\partial \boldsymbol{z}_i} = \frac{\partial L}{\partial \boldsymbol{z}_{i+1}} h_i^{(z)}(\tilde{\boldsymbol{z}}_i, \boldsymbol{\theta}_i), \quad \frac{\partial L}{\partial \boldsymbol{\theta}_i} = \frac{\partial L}{\partial \boldsymbol{z}_{i+1}} h_i^{(\theta)}(\tilde{\boldsymbol{z}}_i, \boldsymbol{\theta}_i), \tag{4}$$

where $\tilde{\boldsymbol{z}}_i$ denotes the pruned activation $\boldsymbol{z}_i$. We inherit the mathematical forms behind $h_i^{(z)}$ and $h_i^{(\theta)}$ from the normal backward to preserve the gradient flow, but calculate the gradients on weights and activations with the pruned layerwise activation $\tilde{\boldsymbol{z}}_i$. With the above backward, there is no need for saving the dense activation $\boldsymbol{z}_i$. Instead, we store a sparse version $\tilde{\boldsymbol{z}}_i$ that is more memory efficient.

The detailed optimization algorithm with the proposed Back Razor can be found in Algorithm 1. In the forward pass, we would compute with dense activation while only saving its sparse version. In the backward, we employ the sparse activation for computing the gradient.

We use a simple pruning method for backward activation pruning: prune the smallest magnitude activations. The smallest magnitude values are usually playing less significant roles than those with larger magnitude. This pruning method is also widely used for many pruning works [35]. In practice, we would prune all the activations to a fixed sparsity of $\lambda$ by setting all the values under $k$ th smallest value as zero, where $k = \lambda n$, $n$ is the total number of values for the activation. As the ranking can be slow for large activations, instead of ranking the whole activation, we rank the activation of each sample and use the sample-wise threshold for pruning. This largely shortens the ranking time. Also, we noted that the thresholds across different samples are very close, which means it is functionally similar to global ranking.

We save the sparse matrix into two parts: a bitmap that indicates the position of the non-zero elements and a smaller dense matrix that contains the values of the non-zero elements. The bitmap has the

---

**Algorithm 1** Backward and Update with Sparse Activation

---

Sample $\boldsymbol{x}, y$ from $\mathcal{D}$
Initialize $\boldsymbol{z}_0 = \boldsymbol{x}$
**for** $i = 1$ to $l$ **do**                                                                                   ▷ Forward pass.
    $\boldsymbol{z}_i = f_i(\boldsymbol{z}_{i-1}, \theta_i)$
    $\tilde{\boldsymbol{z}}_{i-1} \leftarrow \text{TopK\_Prune}(\boldsymbol{z}_{i-1})$                    ▷ Sparsify $\boldsymbol{z}_{i-1}$ and store it.
**end for**
Calculate the training loss $L = \mathcal{L}(\boldsymbol{z}_l, y)$
Initialize $\boldsymbol{s}_l \leftarrow \frac{\partial L}{\partial \boldsymbol{z}_l}$
**for** $i = l - 1$ to $1$ **do**                                                                          ▷ Backward pass.
    Calculate $\frac{\partial L}{\partial \boldsymbol{z}_i}$: $\boldsymbol{s}_i \leftarrow \boldsymbol{s}_{i+1} h_i^{(z)}(\tilde{\boldsymbol{z}}_i, \boldsymbol{\theta}_i)$          ▷ Calculate the gradient for the next activation
    Calculate $\frac{\partial L}{\partial \boldsymbol{\theta}_i}$: $\boldsymbol{s}_{i+1} h_i^{(\theta)}(\tilde{\boldsymbol{z}}_i, \boldsymbol{\theta}_i)$ and update $\boldsymbol{\theta}_i$.
**end for**

---

same shape as the origin sparse tensor, but it is much smaller since it is in bool type. The memory cost of this format is $\frac{n}{8} + \lambda n c_{\text{type}}$ byte, where $c_{\text{type}}$ is the count of bytes for the employed data type.

For the previous activation pruning method, the error that comes from pruning could gradually accumulate layer by layer in the forward, causing the deviation from the pre-trained model and leading to performance degradation. In contrast, the proposed Back Razor ensures the correctness of the forward. Moreover, in the backward progress, the pruning error of many layers would not accumulate: the activations of those layers are only used for computing the gradient of itself (e.g. linear layer, batch/layer normalization layer). It's also worth noting that the proposed Back Razor saves the same amount of memory as activation pruning under the same pruning ratio.

**Comparison with recent parameter-efficient transfer methods.** Recent studies reveal that fine-tuning can be parameter efficient. For example, diffPrune [26] points out that the fine-tuning performance can match fully fine-tuning via only modifying 0.5% of the weights. However, it employs Gumbel-Softmax [36] to make the masked parameters differentiable, which means the dense activations are required for updating the weight. Therefore, it cannot yield memory efficiency. On the other hand, other works also reveal that fine-tuning part of the model can yield competitive performance [27, 28, 29, 37]. Though it can lead to memory saving, we empirically verified that Back Razor can yield a higher saving ratio.

### 3.3 Theoretical Results

Now we study the convergence of Back Razor. We simplify the notion and define $\mathcal{L}$ to be the training loss of a convolutional neural network that takes the model's weights as the input. $\boldsymbol{\theta}_t = \{W_i\}_{i=1}^L$ as the flattened vector of the model's parameters at step $t$, and $\boldsymbol{g}_t$ be the gradient calculated with all the samples in the training dataset. Furthermore, we denote the *stochastic* gradient by $\tilde{\boldsymbol{g}}_t$, which is the gradient calculated with one sample from the training set. We make several classical assumptions [38]:

**Assumption 1.** *Let $\boldsymbol{g}(\boldsymbol{\theta})$ be the gradient of the objective $\mathcal{L}$ at point $\boldsymbol{\theta}$, then for all $\boldsymbol{x}$ and $\boldsymbol{y}$ we assume there exists a non-negative constant vector $\boldsymbol{\beta}$ that*

$$|\mathcal{L}(\boldsymbol{y}) - [\mathcal{L}(\boldsymbol{x}) + g(\boldsymbol{x})(\boldsymbol{y} - \boldsymbol{x})^\top]| \leq \frac{1}{2} \sum_j \boldsymbol{\beta}_j (\boldsymbol{y} - \boldsymbol{x})_j^2$$

**Assumption 2.** *For each iteration $t$, the stochastic gradient $\tilde{\boldsymbol{g}}_t$ satisfies the following conditions:*

$$\mathbb{E}[\tilde{\boldsymbol{g}}_t] = \boldsymbol{g}_t, \qquad \mathbb{E}[(\tilde{\boldsymbol{g}}_t - \boldsymbol{g}_t)_j^2] \leq \sigma_j^2, \forall j \in [n],$$

*where $n$ is the number of parameters of models and $\sigma_j^2$ is a constant representing the variance for coordinate j.*

**Assumption 3.** *The objective $\mathcal{L}$ is bounded below by $L^*$.*

These assumptions are discussed in Appendix. We first prove the convergence for linear networks.

**Lemma 1.** *For linear neural networks, the gradient of parameters can decomposed by $\tilde{\boldsymbol{g}}_t = \tilde{\boldsymbol{z}}_t \tilde{\boldsymbol{A}}_t^\top$, and approximated gradient after activation pruning can decomposed by $\tilde{\boldsymbol{g}}_t' = \tilde{\boldsymbol{z}}_t' \tilde{\boldsymbol{A}}_t^\top$, where $\tilde{\boldsymbol{z}}_t$ is the exact activation and $\tilde{\boldsymbol{z}}_t'$ is the pruned activations.*

In linear networks, $\frac{\partial L}{\partial \boldsymbol{z}_i}$ only depends on the model's weights after the $i$ th layer, so pruning the activation stored for backward does not change the its gradient. Therefore in Lemma 1, $\tilde{\boldsymbol{A}}_t$ is exclusively constructed by activations' gradients, and not changed after activation pruning.

We call $\tilde{\boldsymbol{A}}_t$ the *transformation matrix* for activations at step $t$.

**Theorem 1** (Convergence). *If Assumption 1-3 hold, $(p + \alpha\beta)K_a < 1$ where $p$ is the pruning ratio and $K_a$ is the squared condition number of the transformation matrix, then*

$$\mathbb{E}[\frac{1}{T}\sum_{t=1}^{T}\|\boldsymbol{g}_t\|^2] \leq \frac{2(\mathcal{L}(\boldsymbol{\theta}_0) - L^*)}{[1 - (p + \alpha\beta)K_a]\alpha T} + \frac{(p + \alpha\beta)K_a}{1 - (p + \alpha\beta)K_a}\sigma^2, \tag{5}$$

*where the $T$ is the number of iterations, $\beta = \|\boldsymbol{\beta}\|_\infty$, $\boldsymbol{\theta}_0$ is the initial weights and $\sigma^2 = \sum_{j=1}^{n}\sigma_j^2$.*

The Eqn. 5 consists of two terms: the first term will converge to zero as the number of iterations $T$ goes to infinity. The second term will not vanish, but its value can be controlled by using larger batch size. Intuitively, it proves that our algorithm reach to the neighborhood of a stationary point, and the radius of the neighborhood is bounded by the gradient variance.

**Remark.** The key factor for theoretical convergence is $(p + \alpha\beta)K_a$. $\alpha\beta$ can be reduced by using smaller learning rate. The theoretical convergence will not be guaranteed when the transformation matrix is ill-conditioned.

We next discuss the convergence for non-linear networks, and focus on analyzing convolutional neural networks with a common `CONV-BN-ReLU` structure [15]. The activation pruning happens after the convolutional layer, where the gradient on weights and activations can be calculated as [15]:

$$\frac{\partial L}{\partial \boldsymbol{z}_i} = W_i^\top * \frac{\partial L}{\partial \boldsymbol{z}_{i+1}}, \quad \frac{\partial L}{\partial \boldsymbol{\theta}_i} = \frac{\partial L}{\partial \boldsymbol{z}_{i+1}} * \boldsymbol{z}_i, \tag{6}$$

which suggests the gradient on $\boldsymbol{z}_i$ will not be affected if we use the pruned activation $\tilde{\boldsymbol{z}}_i$ to obtain the derivatives on $\boldsymbol{\theta}_i$ when back-warding through the convolutional layer. The activations are neither stored nor pruned for `BN` and `ReLU` operations, so the Lemma 1 holds for convolutional neural networks. So under the same assumptions, we achieve similar convergence situation for CNNs. We present more details in Appendix.

We lastly discuss the Transformer structures. Unfortunately, the self-attention module does not meet the Lipchitz condition [39], therefore the above theoretical analysis will not apply out-of-the-box to them. However we empirically observe that Back Razor is also applicable on ViTs [40] with decent performance (refer to Section 4.3). A more rigorous theoretical framework customized for transformers is left as our future work.

## 4 Experiment

### 4.1 Settings

Following the common practice [26, 41, 42], we employ supervised pre-trained models on ImageNet as the starting point of transfer learning. Specifically, we employ ImageNet-1K and Imagenet-22K for CNNs and ViTs, respectively. For downstream fine-tuning, we consider eight datasets: Pets [43], Aircraft [44], CIFAR10, CIFAR100 [45], Flowers [46], Cars [47], CUB [48], and Food [49]. The default image resolution is $224 \times 224$ following [1].

Our experiments are implemented with Pytorch [50] and conducted on 1080 Ti or V100 GPUs. To measure the training memory, by default we report the theoretical memory usage following [1]. We also report the actual memory usage measured on the GPU at Section 4.3.

### 4.2 Back Razor with Convolutional Neural Networks

We choose ProxylessNAS-Mobile [52] as the CNN backbone following TinyTL [1]. It is worth noting that we keep the origin setting of ProxylessNAS-Mobile instead of modifying the architecture as in TinyTL given Back Razor is applicable for any architecture. We also follow the most of training settings of TinyTL for fair comparison: The fine-tuning epochs and batch size is set as 50 epochs

Table 1: The comparison between Back Razor with the previous methods on different datasets with CNN. All the reported results are the top1 accuracy (%). FT-Last denotes fine-tuning the linear classifier (the last fully connected layer). FT-Norm+Last denotes fine-tuning the batch normalization layers plus the linear classifier [37, 51]. TinyTL@320 denotes training and inference with a larger resolution of 320. The Back Razor@90% denotes the Back Razor with a sparsity ratio of 90%. The memory footprint of the training at a batch size of 8 (test in CIFAR100) is also reported in the second column. Some experiments are conducted five times with different random seeds. The mean and variance are reported.

| Method | Train Memory | Pets | Aircraft | CIFAR10 | CIFAR100 | Flowers | Cars | CUB | Food |
|---|---|---|---|---|---|---|---|---|---|
| FT-Last | 31MB | 91.3 | 44.9 | 85.9 | 68.8 | 90.1 | 50.9 | 73.3 | 68.7 |
| FT-Norm+Last | 192MB | 92.2 | 68.1 | 94.8 | 80.2 | 94.3 | 77.9 | 76.3 | 77.0 |
| FT-Full | 366MB | 93.0±0.05 | 88.2±0.24 | 97.1±0.02 | 84.1±0.07 | 97.0±0.17 | 91.0±0.06 | 80.9±0.33 | 83.8±0.10 |
| TinyTL [1] | 37MB | 91.8 | 75.4 | 95.9 | 81.4 | 95.5 | 85.0 | 77.1 | 79.7 |
| TinyTL@320 [1] | 65MB | 92.9 | 82.3 | 96.1 | 81.5 | 96.8 | 88.8 | 81.0 | 82.9 |
| Back Razor@90% (ours) | 42MB | 92.9±0.11 | 87.6±0.16 | 96.9±0.07 | 83.3±0.05 | 96.9±0.34 | 90.1±0.10 | 81.0±0.24 | 83.2±0.07 |
| Back Razor@95% (ours) | 41MB | 93.2 | 85.7 | 96.8 | 82.7 | 96.6 | 87.2 | 79.5 | 83.1 |
| Back Razor@97% (ours) | 40MB | 93.0 | 85.2 | 96.6 | 82.6 | 95.9 | 88.1 | 78.8 | 82.3 |
| Back Razor@99% (ours) | 39MB | 92.5 | 80.7 | 96.1 | 80.2 | 95.0 | 85.6 | 76.9 | 80.4 |

and 8, respectively. The model is optimized with adam [53] optimizer and cosine learning learning rate schedule [54]. The initial learning rate is tuned for each dataset. We freeze all the parameters of Batch Normalization layers [55] as they occupy more memory than convolutional layers while having much fewer parameters to update. By freezing them, we could free the occupied memory with less influence on the fine-tuning performance. We also use the origin activation for ReLU6, which is memory efficient (equal to the memory cost of a bitmap with the same shape). The Back Razor is then applied to convolutional layers and the fully-connected classification head.

**Comparison Results:** In Table 1, we compare Back Razor with previous transfer learning methods on CNNs. Specifically, it includes i) FT-Last: fine-tuning the last fully connected layer of the network, which is also referred as linear classification head. ii) FT-Norm+Last: fine-tuning the batch normalization layers plus the classification head iii) FT-Full: fine-tuning the full model. Also, we include two different variants of the previous state-of-the-art (SOTA) method [1]: TinyTL and TinyTL@320 (TinyTL@320 employs a larger resolution image of 320).

Compared with FT-Full, Back Razor@90% use **8.7x** times smaller memory while achieving comparable performance. The variance for BackRazor@90% with respect to different random seeds is also comparable with FT-Full and they are both small. In contrast, FT-Norm+Last takes 4.6x time more memory while being [0.7%, 19.5%, 2.1%, 3.1%, 2.6%, 12.2%, 4.7%, 6.2%] worse than Back Razor@90% in [Pets, Aircraft, CIFAR10, CIFAR100, Flowers, Cars, CUB, Food], respectively. Though FT-Last does yield slightly less training memory (31MB v.s. 42MB), the performance largely degrades: the accuracy is lower by [1.6%, 42.7%, 11.0%, 14.5%, 6.8%, 39.2%, 7.7%, 14.5%] than Back Razor@90% in [Pets, Aircraft, CIFAR10, CIFAR100, Flowers, Cars, CUB, Food], respectively.

Compared with the previous State-of-The-Art method TinyTL, Back Razor@90% is comparable in terms of the training memory usage (37MB v.s 42MB). Meanwhile, it yields higher accuracy of [1.1%, 12.2%, 1.0%, 1.9%, 1.4%, 5.1%, 3.9%, 3.5%] at [Pets, Aircraft, CIFAR10, CIFAR100, Flowers, Cars, CUB, Food], respectively. When compared with TinyTL@320, which employs a larger resolution, Back Razor@90% saves 1.5 times of the memory while still yielding an improvement of [0.0%, 5.3%, 0.8%, 1.8%, 0.1%, 1.3%, 0.0%, 0.3%] at [Pets, Aircraft, CIFAR10, CIFAR100, Flowers, Cars, CUB, Food], respectively.

It's also worth noting that Back Razor method can even achieve more aggressive sparsity. For instance, the performance of Back Razor is [92.9%, 93.2%, 93.0%, 92.5%] for pruning ratio of [90%, 95%, 97%, 99%] at Pets, respectively. The performance maintains the same level even with the pruning ratio of 97% and only marginally drops at the pruning ratio of 99%. A similar trend is also observed in other datasets. However, the high pruning ratio does not bring further memory reduction since the main bottleneck of memory is no longer the accumulated activations. Instead, it is the model parameters (of 11.6MB) and one large forward activation (of 24.5MB).

**Can we prune at forward pass?** We further compare the performance difference between activation pruning (prune both forward and backward) with Back Razor (only prune backward) in Figure 3. We implement the activation pruning via directly applying the mask of the Back Razor to the forward pass. It can be observed that the model can easily collapse (the model can hardly converge and only

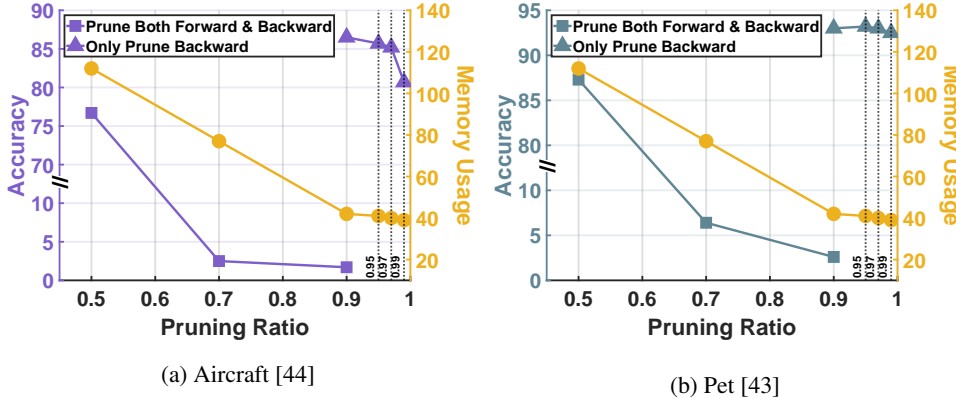

(a) Aircraft [44]

(b) Pet [43]

Figure 3: Comparison between pruning both forward and backward with only pruning backward under different pruning ratios on (a) Aircraft. (b) Pet. For both figures, the y axis in the left is the top 1 accuracy (%) while the y axis in the right is the memory usage, and x-axis is the pruning ratio. The line is coupled with y axis that is with the same color. The yellow line in both figures denotes the memory usage in different pruning ratios. Note that two methods share the same line as their memory consumbtion is the same under the same pruning ratio.

have accuracy below 5%) for activation pruning even with a mild pruning ratio of 70%. In contrast, for Back Razor, the performance can be at a high level even with an aggressive pruning ratio of 99% ( the performance is even higher than activation prune at a sparsity of 50%). This demonstrates that the backward activation can be more sparse than that of the forward.

**Convergence speed in practice:** As shown in Figure 4, the convergence speed in transfer experiments is comparable with each other. In the early stage of the training, the convergence speed of Back Razor is even faster. Even the last stage, FT-Full is only marginally better than Back Razor.

**Comparison with SWAT:** We further compare with SWAT [17] which leverages the similar insight on the sparsity of backward activation. Although [17] was originally developed for training from scratch, here we apply it to the *identical fair setting of transfer learning*. As shown in Table 2, while similar memory footprint is employed for Back Razor@90% and SWAT@80%, the proposed BackRazor can yield a higher performance by 0.6% and 2.6% on CIFAR10 and CIFAR100, respectively.

## 4.3 Back Razor with Vision Transformer

In this section, we study Back Razor on ViT [40]. The patch size is by default set as 16. We employ the standard SGD optimizer with cosine learning rate decay for finetuning. The training steps are fixed as 20k and the initial learning rate is tuned for each dataset. We employ a larger batch size of 128 following the common practice for accelerating training [40, 56, 32].

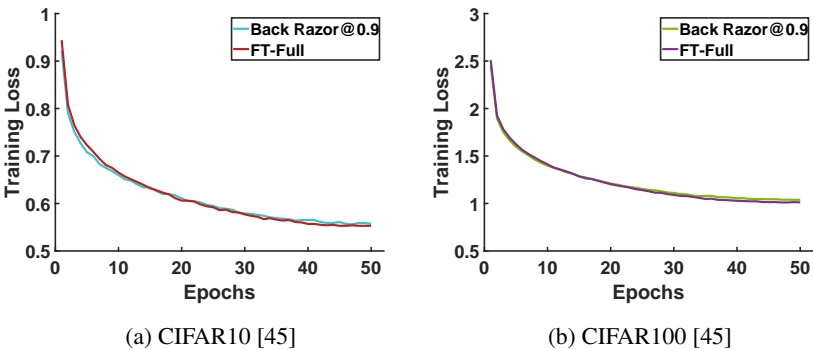

(a) CIFAR10 [45]

(b) CIFAR100 [45]

Figure 4: The training loss convergence speed comparison between Back Razor@0.9 and FT-Full. The convergence speed of them is comparable.

Table 2: Comparison between BackRazor and SWAT [17] on CIFAR10 and CIFAR100 with Resnet-18. The memory footprint of training at a batch size of 128 is reported in the second column. We reproduce SWAT with the official code under fine-tuning setting for fair comparison.

| Method | Train Memory | CIFAR10 | CIFAR100 |
|---|---|---|---|
| FT-Full | 164.4MB | 96.5 | 82.0 |
| SWAT@$80\%$ | 42.7MB | 95.8 | 79.6 |
| SWAT@$90\%$ | 31.7MB | 95.3 | 77.7 |
| Back Razor@$90\%$ (ours) | 41.7MB | 96.4 | 82.2 |

Table 3: The comparison between Back Razor with the previous methods on different datasets with ViT-B/16. All the reported results are the top1 accuracy (%). The memory footprint of the training at a batch size of 128 (compute in CIFAR100) is reported in the second column. Back Razor@$80\%$ + Mesa denotes combining the BackRazor with Mesa.

| Method | Train Memory | Pets | Aircraft | CIFAR10 | CIFAR100 | Flowers | Cars | CUB | Food |
|---|---|---|---|---|---|---|---|---|---|
| FT-Last | 525MB | 91.6 | 44.4 | 96.8 | 86.5 | 99.3 | 57.4 | 85.6 | 86.3 |
| FT-Full | 19235MB | 94.1 | 78.8 | 99.0 | 93.1 | 99.5 | 85.5 | 85.5 | 90.3 |
| Bitfit [27] | 9460MB | 94.0 | 74.2 | 98.8 | 92.3 | 99.5 | 80.6 | 85.7 | 89.7 |
| Mesa [32] | 5442MB | 93.8 | 77.2 | 98.9 | 92.8 | 99.4 | 83.3 | 85.8 | 90.5 |
| Back Razor@$80\%$ (ours) | 4565MB | 93.8 | 77.3 | 98.9 | 92.9 | 99.4 | 84.0 | 86.5 | 90.5 |
| Back Razor@$90\%$ (ours) | 3912MB | 93.3 | 75.7 | 98.9 | 91.8 | 99.4 | 83.4 | 86.6 | 89.8 |
| Back Razor@$95\%$ (ours) | 3496MB | 92.4 | 74.3 | 98.8 | 90.0 | 99.4 | 81.5 | 86.1 | 88.7 |
| Back Razor@$80\%$ + Mesa (ours) | 2937MB | 93.5 | 76.9 | 99.0 | 92.7 | 99.4 | 84.1 | 86.5 | 90.5 |

**Comparison Results:** In Table 3, we compare Back Razor with previous transfer learning methods on ViT, including FT-Last and FT-Full same as defined in Section 4.3. We further compare two parameter efficient transfer learning method from the literature: Bitfit [27], and the SOTA ViT transfer leanring method, Mesa [27] that is based on backpropagation quantization.

Table 4: Comparison of the on-device memory usage on GPU under the batch size of 128 (tested in CIFAR100). We do not report Bitfit here as it cannot yield hardware memory efficiency on GPU with Pytorch.

| Method | On-device Memory |
|---|---|
| FT-Last | 1449MB |
| FT-Full | 22631MB |
| Mesa [32] | 10425MB |
| Back Razor@$80\%$ | 8501MB |
| Back Razor@$90\%$ | 7604MB |
| Back Razor@$95\%$ | 7189MB |
| Back Razor@$80\%$ + Mesa | 6640MB |

Compared with FT-full, Back Razor@$80\%$ yields comparable performance across the datasets while achieving $3.9\times$ memory saving. Though FT-Last employs smaller train memory than Back Razor, the accuracy drops by [2.2%, 32.9%, 2.1%, 6.4%, 0.1%, 26.6%, 0.9%, 4.2%] in [Pets, Aircraft, CIFAR10, CIFAR100, Flowers, Cars, CUB, Food], respectively, compared to Back Razor@$80\%$.

Bitfit achieves competitive performance via tuning less than 0.1% of the parameters. However, it can only reduce the memory saving of $2.0\times$ memory given it requires saving some large activations in full (e.g. attention map). In contrast, Back Razor can reduce the size of any activations. While Mesa has similar memory usage and performance compared with Back Razor, we note that its methodology is orthogonal with Back Razor. By combining Meta and Back Razor together, we can achieve $7.5\times$ memory saving with comparable performance to FT-Full.

When employing a larger pruning ratio for BackRazor in ViTs, more memory footprint savings can be achieved at little performance degradation. Especially, for CIFAR10, the performance of BackRazor only drops by 0.2% compared to FT-Full at a pruning ratio of 95%.

## 4.4 Back Razor with More Tasks

In this section, we further exploring applying the proposed BackRazor to more downstream tasks.

Table 5: BackRazor for Pascal VOC segmentation task with DeepLabV3-MobileNet [57, 58]. The memory footprint of the training at a batch size of 8 is reported in the second row.

| Pruning ratio | 0% (FT-Full) | 10% | 20% | 50% | 70% | 90% | 95% | 97% |
|---|---|---|---|---|---|---|---|---|
| Memory(MB) | 8864 | 8295 | 7467 | 4983 | 3327 | 1671 | 1257 | 1092 |
| Accuracy(%) | 70.9 | 71.0 | 70.9 | 70.6 | 70.5 | 70.3 | 69.4 | 68.5 |

Table 6: BackRazor for RTE from GLUE Benchmark [59] with BERT base [19]. The theoretical and on-device memory footprints of training at a batch size of 8 are reported.

| Method | Theoretical Memory (MB) | Actual GPU Memory (MB) | Accuracy |
|---|---|---|---|
| FT-Full | 4174 | 7002 | 70.4 |
| Back Razor@90% | 2187 | 4858 | 69.7 |

**Semantic Segmentation:** As shown in Table 5, when applying to semantic segmentation tasks, BackRazor can yield $5.3\times$ memory efficiency at a pruning ratio of 90% with only a marginal accuracy drop of 0.6% compared to the fully fine-tuning baseline.

**Language modeling:** As demonstrated in Table 6, when applying Back Razor on the language model, Back Razor can achieve a significant memory efficiency improvement at a pruning ratio of 90% with only a marginal accuracy drop of 0.6%.

## 5   Conclusion

In this work, we propose Back Razor, a memory-efficient transfer learning method, which is also the first asymmetric sparse transfer method that only sparsifies the back-propagation activations while keeping the forward. Through extensive experiments in both Convolutional Neural Networks and Vision Transformer across multiple datasets, we demonstrated the proposed Back Razor can achieve state-of-the-art memory efficiency with an aggressive pruning ratio. Moreover, it can be combined with the propagation quantization method and yield better memory efficiency. As limitations, Back Razor has only been tested on a limited range of tasks, and its theoretical underpinning is so far restricted to CNN backbones despite its empirical success in ViTs, calling for continued efforts.

## Acknowledgment

Z. Wang is in part supported by the National Science Foundation under Grant IIS-2212176, and a Google TensorFlow Model Garden Award.

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
