# Appendix for Back Razor: Memory-Efficient Transfer Learning by Self-Sparsified Backpropagation

**Ziyu Jiang**[†*], **Xuxi Chen**[‡ *], **Xueqin Huang**[†], **Xianzhi Du**[§], **Denny Zhou**[§], **Zhangyang Wang**[‡]
[†]Texas A&M University  [‡]University of Texas at Austin  [§]Google
{jiangziyu,xueq13}@tamu.edu,{xxchen,atlaswang}@utexas.edu
{xianzhi,dennyzhou}@google.com

## A  Theory

### A.1  Assumptions

We provide grounds for assumptions in this section. Assumption 1 is a fine-grained version of the standard Lipschitz smoothness, *i.e.*,

$$|\mathcal{L}(\boldsymbol{y}) - [\mathcal{L}(\boldsymbol{x}) + \boldsymbol{g}(\boldsymbol{x})^\top (\boldsymbol{y} - \boldsymbol{x})]| \leq \frac{\|\beta\|_\infty}{2} \sum_j (\boldsymbol{y} - \boldsymbol{x})_j^2.$$

We follow [1] to make this assumption.

Assumption 2 combines the assumption of the unbiased gradient and bounded variance, which is also standard in literature. The standard variance bound on the gradient can be recovered by defining $\sigma^2 = \sum_j \sigma_j^2$ and we can see that $\mathbb{E}[(\tilde{\boldsymbol{g}}_t - \boldsymbol{g}_t)^2] \leq \sigma^2$.

Assumption 3 defines a lower bound for the objective value, which is necessary for convergence to stationary points.

### A.2  Proof of Theorem 1

Let $L_k$ be training objective at step $k$, *i.e.*, $L_k = \mathcal{L}(\boldsymbol{\theta}_k)$. Let $\boldsymbol{m}$ be the pruning mask for activation $\tilde{\boldsymbol{z}}_k$, then we have $\tilde{\boldsymbol{z}}'_k = \text{diag}(\boldsymbol{m})\tilde{\boldsymbol{z}}_k$.

*Proof.* The difference between the two objectives $L_{k+1} - L_k$ is bounded by:

$$L_{k+1} - L_k \leq \boldsymbol{g}_k(\boldsymbol{\theta}_{k+1} - \boldsymbol{\theta}_k)^\top + \sum_j \frac{\beta_j}{2}(\boldsymbol{\theta}_{k+1} - \boldsymbol{\theta}_k)_j^2 \qquad (Assumption\ 1)$$

$$\leq \boldsymbol{g}_k(\boldsymbol{\theta}_{k+1} - \boldsymbol{\theta}_k)^\top + \frac{\|\boldsymbol{\beta}\|_\infty}{2}\|\boldsymbol{\theta}_{k+1} - \boldsymbol{\theta}_k\|_2^2$$

$$= -\alpha_k \boldsymbol{g}_k \tilde{\boldsymbol{g}}'^\top_k + \alpha_k^2 \frac{\beta}{2}\|\tilde{\boldsymbol{g}}'_k\|_2^2,$$

which is based on the SGD update rule ( $\boldsymbol{\theta}_{k+1} - \boldsymbol{\theta}_k = -\alpha_k \tilde{\boldsymbol{g}}'_k$) where $\tilde{\boldsymbol{g}}_k$ represents the pruned stochastic gradient.

For the first term, we have

$$-\boldsymbol{g}_k \tilde{\boldsymbol{g}}'^\top_k = -\boldsymbol{g}_k \tilde{\boldsymbol{g}}'^\top_k + \boldsymbol{g}_k(\tilde{\boldsymbol{g}}_k - \tilde{\boldsymbol{g}}'_k)^\top,$$

---

[*]Equal contribution

36th Conference on Neural Information Processing Systems (NeurIPS 2022).

and

$$
\begin{aligned}
\boldsymbol{g}_k(\tilde{\boldsymbol{g}}_k - \tilde{\boldsymbol{g}}'_k)^\top &= \boldsymbol{g}_k \tilde{\boldsymbol{A}}_k (\tilde{\boldsymbol{z}}_k - \tilde{\boldsymbol{z}}'_k)^\top \\
&= \boldsymbol{g}_k \tilde{\boldsymbol{A}}_k (I - \mathrm{diag}(\boldsymbol{m})) \tilde{\boldsymbol{z}}_k^\top \\
&= \left\langle \boldsymbol{g}_k, \tilde{\boldsymbol{A}}_k (I - \mathrm{diag}(\boldsymbol{m})) \tilde{\boldsymbol{z}}_k^\top \right\rangle \\
&\leq \frac{1}{2} (\|\boldsymbol{g}_k\|_2^2 + \|\tilde{\boldsymbol{A}}_k (I - \mathrm{diag}(\boldsymbol{m})) \tilde{\boldsymbol{z}}_k^\top\|_2^2) \\
&= \frac{1}{2} (\|\boldsymbol{g}_k\|_2^2 + \tilde{\boldsymbol{z}}_k (I - \mathrm{diag}(\boldsymbol{m})) \tilde{\boldsymbol{A}}_k^\top \tilde{\boldsymbol{A}}_k (I - \mathrm{diag}(\boldsymbol{m})) \tilde{\boldsymbol{z}}_k^\top)
\end{aligned}
$$

Assume $\tilde{\boldsymbol{A}}_k$ has positive singular values, we have

$$
\tilde{\boldsymbol{z}}_k (I - \mathrm{diag}(\boldsymbol{m})) \tilde{\boldsymbol{A}}_k^\top \tilde{\boldsymbol{A}}_k (I - \mathrm{diag}(\boldsymbol{m})) \tilde{\boldsymbol{z}}_k^\top = \frac{\tilde{\boldsymbol{z}}_k (I - \mathrm{diag}(\boldsymbol{m})) \tilde{\boldsymbol{A}}_k^\top \tilde{\boldsymbol{A}}_k (I - \mathrm{diag}(\boldsymbol{m})) \tilde{\boldsymbol{z}}_k^\top}{\tilde{\boldsymbol{z}}_k \tilde{\boldsymbol{A}}_k^\top \tilde{\boldsymbol{A}}_k \tilde{\boldsymbol{z}}_k^\top} \tilde{\boldsymbol{z}}_k \tilde{\boldsymbol{A}}_k^\top \tilde{\boldsymbol{A}}_k \tilde{\boldsymbol{z}}_k^\top
$$

$$
\leq \frac{\tilde{\boldsymbol{z}}_k (I - \mathrm{diag}(\boldsymbol{m})) \tilde{\boldsymbol{A}}_k^\top \tilde{\boldsymbol{A}}_k (I - \mathrm{diag}(\boldsymbol{m})) \tilde{\boldsymbol{z}}_k^\top}{\|\tilde{\boldsymbol{z}}_k (I - \mathrm{diag}(\boldsymbol{m}))\|_2^2} \frac{\|\tilde{\boldsymbol{z}}_k (I - \mathrm{diag}(\boldsymbol{m}))\|_2^2}{\|\tilde{\boldsymbol{z}}_k\|_2^2} \frac{\|\tilde{\boldsymbol{z}}_k\|_2^2}{\tilde{\boldsymbol{z}}_k \tilde{\boldsymbol{A}}_k^\top \tilde{\boldsymbol{A}}_k \tilde{\boldsymbol{z}}_k^\top} \tilde{\boldsymbol{z}}_k \tilde{\boldsymbol{A}}_k^\top \tilde{\boldsymbol{A}}_k \tilde{\boldsymbol{z}}_k^\top
$$

$$
\leq \frac{n - K}{n} \frac{\max \lambda^2(\tilde{\boldsymbol{A}}_k)}{\min \lambda^2(\tilde{\boldsymbol{A}}_k)} \tilde{\boldsymbol{z}}_k \tilde{\boldsymbol{A}}_k^\top \tilde{\boldsymbol{A}}_k \tilde{\boldsymbol{z}}_k^\top
$$

$$
= pK_a \tilde{\boldsymbol{z}}_k \tilde{\boldsymbol{A}}_k^\top \tilde{\boldsymbol{A}}_k \tilde{\boldsymbol{z}}_k^\top
$$

$$
= pK_a \|\tilde{\boldsymbol{g}}_k\|^2
$$

where $p := \frac{n-K}{n}$ is the pruning ratio and $K_a := \frac{\max \lambda^2(\tilde{\boldsymbol{A}}_k)}{\min \lambda^2(\tilde{\boldsymbol{A}}_k)}$ is the squared condition number of $\tilde{\boldsymbol{A}}_k$. This is because

1.
$$
\frac{\tilde{\boldsymbol{z}}_k (I - \mathrm{diag}(\boldsymbol{m})) \tilde{\boldsymbol{A}}_k^\top \tilde{\boldsymbol{A}}_k (I - \mathrm{diag}(\boldsymbol{m})) \tilde{\boldsymbol{z}}_k^\top}{\|\tilde{\boldsymbol{z}}_k (I - \mathrm{diag}(\boldsymbol{m}))\|_2^2} \leq \max \lambda^2(\tilde{\boldsymbol{A}}_k)
$$

2.
$$
\frac{\|\tilde{\boldsymbol{z}}_k (I - \mathrm{diag}(\boldsymbol{m}))\|_2^2}{\|\tilde{\boldsymbol{z}}_k\|_2^2} \leq \frac{n - K}{n}
$$
since $\boldsymbol{m}$ is generated by the TopK operation.

3.
$$
\frac{\|\tilde{\boldsymbol{z}}_k\|_2^2}{\tilde{\boldsymbol{z}}_k \tilde{\boldsymbol{A}}_k^\top \tilde{\boldsymbol{A}}_k \tilde{\boldsymbol{z}}_k^\top} \leq \frac{1}{\min \lambda^2(\tilde{\boldsymbol{A}}_k)}
$$

The second term can be bounded by

$$
\begin{aligned}
\|\tilde{\boldsymbol{g}}'_k\|_2^2 &= (\tilde{\boldsymbol{z}}_k \mathrm{diag}(\boldsymbol{m}) \tilde{\boldsymbol{A}}_k^\top)(\tilde{\boldsymbol{z}}_k \mathrm{diag}(\boldsymbol{m}) \tilde{\boldsymbol{A}}_k^\top)^\top \\
&= \tilde{\boldsymbol{z}}_k \mathrm{diag}(\boldsymbol{m}) \tilde{\boldsymbol{A}}_k^\top \tilde{\boldsymbol{A}}_k \mathrm{diag}(\boldsymbol{m}) \tilde{\boldsymbol{z}}_k^\top \\
&= \frac{\tilde{\boldsymbol{z}}_k \mathrm{diag}(\boldsymbol{m}) \tilde{\boldsymbol{A}}_k^\top \tilde{\boldsymbol{A}}_k \mathrm{diag}(\boldsymbol{m}) \tilde{\boldsymbol{z}}_k^\top}{\tilde{\boldsymbol{z}}_k \tilde{\boldsymbol{A}}_k^\top \tilde{\boldsymbol{A}}_k \tilde{\boldsymbol{z}}_k^\top} \tilde{\boldsymbol{z}}_k \tilde{\boldsymbol{A}}_k^\top \tilde{\boldsymbol{A}}_k \tilde{\boldsymbol{z}}_k^\top \\
&= \frac{\tilde{\boldsymbol{z}}_k \mathrm{diag}(\boldsymbol{m}) \tilde{\boldsymbol{A}}_k^\top \tilde{\boldsymbol{A}}_k \mathrm{diag}(\boldsymbol{m}) \tilde{\boldsymbol{z}}_k^\top}{\|\tilde{\boldsymbol{z}}_k \mathrm{diag}(\boldsymbol{m})\|_2^2} \frac{\|\tilde{\boldsymbol{z}}_k \mathrm{diag}(\boldsymbol{m})\|_2^2}{\|\tilde{\boldsymbol{z}}_k\|_2^2} \frac{\|\tilde{\boldsymbol{z}}_k\|_2^2}{\tilde{\boldsymbol{z}}_k \tilde{\boldsymbol{A}}_k^\top \tilde{\boldsymbol{A}}_k \tilde{\boldsymbol{z}}_k^\top} \tilde{\boldsymbol{z}}_k \tilde{\boldsymbol{A}}_k^\top \tilde{\boldsymbol{A}}_k \tilde{\boldsymbol{z}}_k^\top \\
&\leq K_a \|\tilde{\boldsymbol{g}}_k\|_2^2.
\end{aligned}
$$

Table 1: The comparison between Back Razor with the previous methods on CUB200 and Flowers with ViT-L/16. All the reported results are the top1 accuracy (%). The memory footprint of the training at a batch size of 128 (compute in CIFAR100) is reported in the second column.

| Method | Train Memory | CUB200 | Flowers |
|---|---|---|---|
| FT-Full | 51257.0MB | 86.6 | 99.6 |
| Back Razor@90% (ours) | 12270.4MB | 86.9 | 99.5 |

Table 2: The comparison between Back Razor and ActNN [2] on CUB200 and Flowers with ProxylessNAS-Mobile. All the reported results are the top1 accuracy (%). The memory footprint of the training at the batch size of 8 (compute in CIFAR100) is reported in the second column.

| Method | Train Memory | CUB200 | Flowers |
|---|---|---|---|
| ActNN [2] | 60MB | 81.0 | 96.9 |
| Back Razor@90% (ours) | 42MB | 81.0 | 96.9 |

Combining these two items together:

$$L_{k+1} - L_k \leq \alpha_k(-\boldsymbol{g}_k^\top \tilde{\boldsymbol{g}}_k + \frac{1}{2}\|\boldsymbol{g}_k\|_2^2 + \frac{1}{2}pK_a\|\tilde{\boldsymbol{g}}_k\|^2) + \frac{\alpha_k^2 \beta}{2}K_a\|\tilde{\boldsymbol{g}}_k\|_2^2$$

Next we find the expected improvement at time $k+1$:

$$\mathbb{E}[L_{k+1} - L_k|\boldsymbol{\theta}_k] \leq -\alpha_k\frac{1}{2}\|\boldsymbol{g}_k\|_2^2 + \alpha_k\frac{1}{2}pK_a(\|\boldsymbol{g}_k\|_2^2 + \sigma^2) + \frac{\alpha_k^2 \beta}{2}K_a(\|\boldsymbol{g}_k\|_2^2 + \sigma^2),$$

and by extending the expectation over the trajectory and summing over $T$ iterations we finish the proof. $\square$

## B  More Experiment Results

### B.1  Back Razor on Larger Pre-trained Models

We test Back Razor on larger pre-trained model ViT-L/16. As shown in Table 1, Back Razor can achieve comparable performance with FT-Full for both CUB200 and Flowers datasets. Remarkably, it can even surpass CUB200 while being more memory efficient.

### B.2  Compared with ActNN

We further compare Back Razor with the ActNN [2], which conducts the backpropagation quantization for CNN. As shown in Table 2, though the ActNN yields similar performance with Back Razor, Back Razor requires less memory footprint.

### B.3  Small Batch Size

We further study if Back Razor works on extreme small batch size of 1. As shown in Table 3, Back Razor can achieve higher performance with less memory usage compared to TinyTL under a batch size of 1.

Table 5: The on-device memory comparison between Back Razor and fully fine-tuning (test at a batch size of 8 in CIFAR100).

| Batch size | Baseline | Back Razor@90% |
|---|---|---|
| 8 | 1655MB | 1637MB |
| 128 | 7393MB | 6634MB |

Table 3: The comparison between Back Razor and TinyTL on Flowers with batch size of 1. The model is ProxylessNAS.

| Method | Train Memory | Flowers |
|---|---|---|
| FT-Full | 34MB | 96.4 |
| TinyTL | 18MB | 96.1 |
| Back Razor@80% (ours) | 15MB | 96.4 |

Table 4: The comparison between Back Razor and fully fine-tuning on Flowers with SGD. The model is ProxylessNAS.

| Method | Train Memory | Flowers |
|---|---|---|
| FT-Full | 366MB | 97.1 |
| Back Razor@80% (ours) | 42MB | 96.7 |

## B.4 Different Optimizer

We further study if Back Razor works with SGD optimizer for ProxylessNAS. As shown in Table 4, Back Razor can achieve comparable performance with less memory usage compared to fully fine-tuning with SGD optimizer.

## B.5 On-device Memory Usage in the Case of the CNN Architecture

The on-device (GPU) memory usage of ProxylessNAS-Mobile is illustrated in the Table 5. With a batch size of 8, the memory of Back Razor@90% is comparable with the baseline. For a larger batch size of 128, Back Razor@90% can save 759MB of memory compared to baseline. There is still space for optimizing the implementation as it does not achieve the theoretical performance.