# OpenReview forum: "Back Razor: Memory-Efficient Transfer Learning by Self-Sparsified Backpropagation"
_NeurIPS.cc/2022/Conference — NeurIPS 2022 Accept_

### Official Review · Reviewer_3voy · 2022-07-07

**Rating:** 6
**Confidence:** 4
**Soundness:** 3 good
**Presentation:** 3 good
**Contribution:** 3 good

**Summary:**

Back Razor is a memory-efficient algorithm for transfer learning on edge devices, which works by using sparse activations during the backward pass of neural network training. The paper supports its claims with experimental validation on four image classification datasets and two architectures (CNN and ViT), as well as some theoretical convergence results.


**Questions:**

The main suggestion to the authors is to conduct a substantially more thorough experimental validation of the method.

**Limitations:**

The authors correctly pointed out that the paper only looks at image classification tasks, and also addressed the limitations of their theory’s application to ViTs.

**Strengths And Weaknesses:**

## Strengths

The proposed method is elegant, simple to understand, and easy to implement. If the early experimental results hold up, this method can substantially help with neural network training on edge devices. Experimental evaluation looks quite promising.

The placement of the algorithm in the context of transfer learning on edge devices is appropriate and compelling. Overall, the paper flows well.

## Weaknesses

### [major issue] Experiments [This was addressed to my satisfaction during the discussion period]
The authors only show experiments for vision tasks using two architectures (a CNN optimized for mobile performance and ViT). Besides the basic baselines of full finetuning / FC finetuning / FC+Norm, the paper only benchmarks on one other method in the case of CNNs and two methods in the case of ViTs; it is not clear from the paper that these methods are SOTA for each of the tasks considered. In almost all cases, the experiments are presumably run only once, as generally no confidence intervals were given.

The biggest concern is that the experimental coverage is insufficient to draw strong conclusions. In particular, TinyTL, which is used as a benchmark, provides results for nine datasets of which BackRazor is only tested on four for CNN, and three for ViT. There is also no investigation of how well the method performs with different batch sizes and optimizers, or for other categories of tasks, such as segmentation or language understanding. In the case of CNNs, it would also be useful to see a comparison to other methods that use reduced precision (i.e. quantization); for ViT this was provided.

Finally, it would be useful to see the actual on-device memory usage in the case of the CNN architecture.

### [minor issue] Theory [This was addressed to my satisfaction during the discussion period]
The theory gives a convergence result for a linear network, and then some argumentation is made to extend this result to CNNs. However, Theorem 1 states that the algorithm converges to a neighborhood of a stationary point, and the argument that this neighborhood can be reduced by increasing the batch size does not work in this memory-constrained setting. Thus, it is not clear how well the theory applies, even though it is also experimentally verified that the convergence rates are similar, hence this is overall a minor issue.

### [moderate issue] Writing [This was addressed to my satisfaction during the discussion period]
While the flow of the paper is good, important details are often not made as clear as they could be, or are hard to find. For example, the TopK operator used to prune the activations should be clearly indicated in the algorithm. The theoretical assumptions, and the grounds for them are not explained, leaving the reader to consult another paper for context. Finally, the transition in line 129 of moving from an arbitrary activation $z_i$ to the final activation $z_l$ is somewhat confusing, since none of the rest of the paper focuses on the last layer specifically.

Some sentences make overly broad claims, for instance the very first statement of the abstract calls transfer learning the unqualified _de facto_ choice, which is not generally true. Likewise, in line 101, the claim “transfer learning always happens on personal computing devices” is unsubstantiated.

The proof in the appendix contains numerous typos in the math notation, for instance:
In line 4, $ \tilde{z}’_{k} = \text{diag}(m)(\tilde{z}$ should have been  $ \tilde{z}’_{k} = \text{diag}(m)(\tilde{z}_{k}$
In line 6, $\tilde{g}_k$ should have been $\tilde{g}_k$
In the last formula of line 5, $\beta_i$ should have been the boldface $\beta$. Also, some of the gradients in that equation are incorrectly marked as the full gradient $g$ rather than $\tilde{g}$.
The $\kappa$s in line 13 should have been $K_a$.

Other typos are likewise numerous, although they generally do not adversely affect understanding:

Line 40 of -> or
Line 99 - missing “the” before “de facto”
Line 102 constrain -> constraints
Line 124 by -> of
Line 130 FLOPs should be capitalized
Line 157 actions -> activations
Line 166 value -> values
Line 179 weight -> weights
Line 201 Assumption -> assumptions
Line 231 the word “based” shouldn’t be there
Line 267 Missing verb after “even”
Line 288 employ -> employs
Line 313 illustrate -> illustrates
Line 329 backpropogation -> backpropagation

---

> ### Author Response · Authors · 2022-08-02
> **Response to reviewer 3voy (1/2)**
>
> ## Experiments
> ### 1. baseline, architecture choosing, and error bar
>
> * We argue that the chosen benchmarks are representative and at the SOTA level. The chosen two networks are representative of the mobile CNN(Convolutional Neural Networks) (ProxylessNAS-Mobile) and the Vision Transformers (ViT-B/16). They are also very different from each other: one is based on convolutional layers and the other is based on attention mechanism. Good performance on these two networks indicates the proposed Back Razor can work on a wide range of networks.
> * For the benchmark choosing, we follow the benchmark of TinyTL [1] (Neurips, 2020) for ProxylessNAS-Mobile. We compare with BitFit [2] (ACL, 2022) and [3] (Arxiv, 2022) for ViT-B/16. All of them are recent publications. Two of these works are also accepted to top conferences. We believe this demonstrates the chosen baselines are at SOTA level,
> * We offer the error bar for Back Razor@90% and FT-Full in Table 1 by running five times with different random seeds. The proposed Back Razor is very robust with a small standard deviation of [0.11, 0.16, 0.07, 0.05] for [Pets, Aircraft, CIFAR10, CIFAR100], respectively.
>
> ### 2. Experimental coverage
> We want to refer you to the common response, where we add more empirical results. Specifically, we conduct experiments on four more datasets for ProxylessNAS-Mobile and five more datasets for ViT-B/16. The CelebA experiment requires the pre-train model on VGGFace2. As we cannot finish the pre-train in the short rebuttal window, we skip the experiment on this dataset. Moreover, we also explore different batch size settings and a new optimizer choice:
>
> To verify the proposed Back Razor can work for different batch size. We follow TinyTL conducting experiments with a batch size of 1 for ProxylessNAS-Mobile. As shown in the following table. The proposed BackRazor can achieve higher performance with less memory usage compared to TinyTL under a batch size of 1.
> | Method                | Training Memory | Flowers  |
> |-----------------------|-----------------|----------|
> | FT-Full               | 34MB            | 96.4%    |
> | TinyTL                | 18M             | 96.1%    |
> | Back Razor@80% (ours) | 15MB            | 96.4%    |
>
>
> To verify the proposed Back Razor can work for different optimizers. We conduct experiments for ProxylessNAS-Mobile on  SGD optimizer (the previous optimizer is adam). As shown in the following table, the proposed Back Razor can achieve comparable performance with less memory usage compared to fully fine-tuning with SGD optimizer.
> | Method                | Training Memory | Flowers  |
> |-----------------------|-----------------|----------|
> | FT-Full               | 366MB           | 97.1%    |
> | Back Razor@80% (ours) | 42MB            | 96.7%    |
>
> For other experiments (including the experiments on more tasks as well as the quantization of CNN baselines), we promise to add these experiments in the future version given that the rebuttal window is short.
>
> ### 3. On-device memory usage in the case of the CNN architecture
>
> The on-device (GPU) memory usage of ProxylessNAS-Mobile is illustrated in the following Table. With a batch size of 8, the memory of Back Razor@90% is comparable with the baseline. For a larger batch size of 128, Back Razor@90% can save 759MB of memory compared to baseline. There is still space for optimizing the implementation as it does not achieve the theoretical performance.
>
> | batch size | baseline | Back Razor@90% |
> |------------|----------|----------------|
> | 8          | 1655MB   | 1637MB         |
> | 128        | 7393MB   | 6634MB         |
>
>
> ## Theory
> ### 1. Convergence concern
>
>  We clarify that Theorem 1 gives only the convergence guarantees of Back Razor for MLPs and CNNs but not the quality of models. As we have observed that convergence speeds are similar, we believe that our theory is well-applied in our cases. In the view of experiments, we show the performance with batch sizes up to 128 and have demonstrated superior performance under the memory-constrained setting in our manuscript, and we believe such batch size can be considered as ``large’’ so our method is working. We are open to clarifying any further confusion on this point.
>
> [r1] ON LARGE-BATCH TRAINING FOR DEEP LEARNING: GENERALIZATION GAP AND SHARP MINIMA

---

> > ### Author Response · Authors · 2022-08-02
> > **Response to reviewer 3voy (2/2)**
> >
> > ## Writing
> > ### 1. Details
> > We have followed your suggestion to add a clear description on the `TopK` operator, the theoretical assumptions and reasoning in the revision, and we have revised the mathematical notion on $z_l$ to make it more clear.
> >
> > ### 2. Broad Claims
> > We admit the de facto choice is too strong a claim. We have rephrased the claim in the draft by changing “de facto choice” to “widely used”. There is a typo on  “transfer learning always happens on personal computing devices”, it should be “many transfer learning happen on personal computing devices”. However, these are not the main claim of the paper.
> >
> > ### 3. Typos
> > Thank you for your suggestion. We have proofread our manuscript again and done a revision, which has been highlighted with blue text in the new draft.

---

> > > ### Comment · Reviewer_3voy · 2022-08-03
> > > **Will the new experimental results be added to the manuscript?**
> > >
> > > I appreciate the authors' diligence in responding to my and other reviewers' concerns and running additional dataset experiments. I will respond to the rebuttal in more detail. However, my primary concern was the lack of experimental evidence in the paper, and it seems that the new experimental results were not added to the updated manuscript. I respectfully request the authors to do so.

---

> > > > ### Author Response · Authors · 2022-08-03
> > > > **Response to reviewer 3voy -- add new results**
> > > >
> > > > Thanks for the quick response. The new experimental results have been added to the manuscript and appendix (marked in blue). We are running the missing baselines and would add them once done.

---

> > > > > ### Comment · Reviewer_3voy · 2022-08-04
> > > > > **Thank you for the revisions**
> > > > >
> > > > > Dear paper authors,
> > > > >
> > > > > Thank you for adding the new results and incorporating other feedback. Since this fixes my major concern about the paper, I will raise my score.

---

> > > > > > ### Author Response · Authors · 2022-08-05
> > > > > > **Thank you for raising score**
> > > > > >
> > > > > > Thank you very much for raising the score!

---

### Official Review · Reviewer_RTcC · 2022-07-11

**Rating:** 5
**Confidence:** 5
**Soundness:** 3 good
**Presentation:** 4 excellent
**Contribution:** 3 good

**Summary:**

To reduce memory use of training a transfer learning model, the paper proposes to prune activations stored for back-propagation, instead of traditionally pruning activations stored for both forward-propagation and back-propagation. Experiments on CNN and ViT show that the proposed method achieves comparable accuracy with more aggressive memory reduction compared to competitive methods TinyTL, Bitfit, and Mesa.

**Questions:**

1) Is it possible to provide peak memory use during training?
2) Have you tried structural sparsification strategies?
3) Is it possible to qualitatively or quantitatively compared with the traditional method gradient checkpointing in the experiments?


**Limitations:**

As I mentioned above, a major limitation is the adopted unstructural sparsification strategy which incurs discrepancy between theoretical memory use and practical memory use.

**Strengths And Weaknesses:**

Strengths.
1) In Section 1 and Section 3.1, the paper provides clear explanations on why the paper focuses on compressing activations stored for back-propagation.
2) In Section 3.2, the proposed techniques are clearly written and reasonable. Theoretical analysis on convergence is also provided in Section 3.3.
3) Experiments on CNN and ViT are convincing. For ViT, the paper provides comparison between theoretical memory use and practical memory use in Table 2 and Table 3, respectively.

Weaknesses.
1) I think the authors should provide peak memory use in the experiments. Since the method prunes activations stored for back-propagation, I agree with the authors that the theoretical memory use and averaged memory use are both decreased. However, as the activations are not sparsified during forward-propagation, I am not sure whether peak memory use is decreased compared to previous methods. If the peak memory use is not decreased, the proposed method actually requires computing devices with the same memory capacity as previous methods.
2) A potential problem of the method is discrepancy between theoretical memory use and practical memory use, as shown in Table 2 and Table 3. The problem is incurred by the adopted unstructural sparsification strategy, pruning the smallest magnitude activations, as introduced in Section 3.2. According to memory working mechanism, memory is accessed in column-wise or row-wise manner. A column/row is accessed even if there is only one non-zero element. Therefore, unstructural sparsification cannot effectively practical memory use.
3) I think the traditional method gradient checkpointing is very related to this paper. Is it possible to qualitatively or quantitatively compared with this method in the experiments?

---

> ### Author Response · Authors · 2022-08-02
> **Response to Reviewer RTcC**
>
> ### 1. Concerns about the memory usage computing
>
> * To clarify, the memory usage reported in the paper is all the peak memory. We totally agree with the reviewer that peak memory determines the memory requirement for computing devices.
>
> * It is worth noting that the activations are sparsified during the forward propagation. As shown in Algorithm 1, in the forward pass, we would prune and save the sparsified activation $\tilde{z}_{i-1} $ each time it finishes the forward on layer $f_i$.
> In contrast, the traditional algorithm requires saving the full precision activation.
>
> ### 2. The discrepancy between theoretical memory use and practical memory use
>
> Good question. Directly sparsifying the activation in the original format cannot lead to memory saving. To address this, we structuralize the pruned activation before saving it. As described in lines 165-169, the sparse matrix would be saved with a bitmap and a much smaller dense matrix. This makes savings of sparse activation memory efficient.
>
> ### 3. Comparison with gradient checkpointing
>
> Thanks for pointing out this relevant work. When applying the gradient checkpointing on the ProxylessNAS-Mobile, it can achieve a memory usage of 96.1MB with the same accuracy as the fine-tune baseline (as it wouldn’t change the backward results). In comparison. Our method can be more memory efficient with 42MB memory (Back Razor@90%) with comparable accuracy (as shown in Table 1). Moreover, our method requires no extra computational cost while gradient checkpointing requires one more forward pass. Also, it is worth noting that the proposed method can be combined with gradient checkpointing to further improve memory efficiency.

---

> > ### Author Response · Authors · 2022-08-05
> > **Look forward to further feedback**
> >
> > Dear Reviewer RTcC,
> >
> > Since the discussion section has been started for a few days, it would be highly appreciated if you could look at the above responses and reply. In this way, if you still have concerns, we could have time to address them before due of the discussion section.
> >
> > We would also highly appreciate it if you could consider raising the score if our response has addressed your concerns.
> >
> > Thank you very much for your time and efforts.

---

> > ### Comment · Reviewer_RTcC · 2022-08-05
> > **About your structuralization step**
> >
> > Thanks for your response. You have addressed most of my concerns. Based on your response, the proposed method adopts unstructured activation spasification and then structuralize the pruned activation. My suggestion is that you can directly use structured sparsification methods, the structuralization time can thus be saved. Please correct me if I misunderstood your structuralization step.

---

> > > ### Author Response · Authors · 2022-08-06
> > > **Response to Reviewer RTcC**
> > >
> > > Thanks for your response. We conduct new experiments by adapting the channel-wise structural sparsification on BackRazor with ProxylessNAS-Mobile. As shown in the following Table, in Flowers, compared to the unstructural Back Razor@90%, the accuracy drops for the structural Back Razor@90%. However, the structural Back Razor@80% can still surpass TinyTL@320 with higher accuracy and lower memory usage.
> > >
> > > | Method                        | Training Memory | Flowers  |
> > > |-------------------------------|-----------------|----------|
> > > | FT-Last                       | 31MB            | 90.1     |
> > > | FT-Norm+Last                  | 192MB           | 94.3     |
> > > | FT-Full                       | 366MB           | 96.8     |
> > > | TinyTL                        | 37MB            | 95.5     |
> > > | TinyTL@320                    | 65MB            | 96.8     |
> > > | Back Razor@90% (unstructural) | 42MB            | 97.1     |
> > > | Back Razor@80% (structural)   | 56MB            | 96.9     |
> > > | Back Razor@90% (structural)   | 42MB            | 96.2     |

---

### Official Review · Reviewer_i5p7 · 2022-07-11

**Rating:** 5
**Confidence:** 2
**Soundness:** 3 good
**Presentation:** 2 fair
**Contribution:** 3 good

**Summary:**

The paper proposes a simple and memory-efficient method called Back Razor that prunes activations only for back-propagation, presents a theoretical analysis that the convergence rate of Back Razor can be similar to that of SGD, and shows the effectiveness of Back Razor on CNN and Vision Transformer.

**Questions:**

In Figure 3, it would be better if the memory footprint of training as well as accuracy is plotted when comparing pruning both forward and backward with pruning only backward.

**Limitations:**

As mentioned in the paper, the authors plan to apply Back Razor to several downstream tasks.

**Strengths And Weaknesses:**

Strengths

(1) The paper is well structured and the main idea is clearly explained.

(2) The extensive experiments demonstrate the efficacy of Back Razor to some extent.

Weaknesses

(1) Proof-reading is required due to some errata and awkward expressions.

(2) More experiments on large models such as ViT-L/16 seem to be needed to verify whether Back Razor is really working well on large pre-trained models.

---

> ### Author Response · Authors · 2022-08-02
> **Response to Reviewer i5p7**
>
> ### 1. Proof-reading
> Thanks for pointing it out. We have proofread our manuscript again and done a revision, which has been highlighted with blue text in the new draft.
>
> ### 2. More experiments on large models such as ViT-L/16
>
> Great suggestion. The comparison between the proposed Back Razor and fully fine-tune baseline on ViT-L/16 is illustrated in the following table (employ training setting). The proposed Back Razor can achieve comparable performance with FT-Full for both CUB200 and Flowers datasets. Remarkably, it can even surpass CUB200 while being more memory efficient.
> | dataset        | Memory    | CUB200 | Flowers |
> |----------------|-----------|--------|---------|
> | FT-Full        | 51257.0MB | 86.6%  | 99.6%   |
> | Back Razor@90% | 12270.4MB | 86.9%  | 99.5%   |
>
>
> ### 3. Plot the memory footprint of training
>
> This is a nice suggestion. We have added this to our draft.

---

> > ### Author Response · Authors · 2022-08-05
> > **Look forward to further feedback**
> >
> > Dear Reviewer i5p7,
> >
> > Since the discussion section has been started for a few days, it would be highly appreciated if you could look at the above responses and reply. In this way, if you still have concerns, we could have time to address them before due of the discussion section.
> >
> > We would also highly appreciate it if you could consider raising the score if our response has addressed your concerns.
> >
> > Thank you very much for your time and efforts.

---

> > > ### Comment · Reviewer_i5p7 · 2022-08-09
> > > **Response to authors' feedback**
> > >
> > > I appreciate your reply to address my concerns.  Although there is no experimental results about NLP, authors answered all my questions thoroughly. So, I maintain my score to acceptance.

---

### Official Review · Reviewer_agR8 · 2022-07-20

**Rating:** 6
**Confidence:** 3
**Soundness:** 2 fair
**Presentation:** 2 fair
**Contribution:** 2 fair

**Summary:**

* This paper considers the scenario that one want to not only deploy pre-trained models but also continue train/finetune these models on local devices. It proposes a novel method called Black Razor that reduces the training/finetuning memory of neural networks, by pruning/compressing the activation during back-propagation. BlackRazor is able to achieve 96% sparsity, saving 9.2x memory without losing accuracy. The paper also conducts a theoretical analysis of BlackRazor's convergence.

**Questions:**

* Typo in Figure 2: “Storeage”
* What is the memory difference between activations in forward pass and backward pass? How can pruning activations in backpropagation reduce memory by more than 2x?
* Have you tried on language transformers?

**Limitations:**

N'/'A

**Strengths And Weaknesses:**

Strength:
* The proposed method seems to have great results: successfully reducing much training memory without much loss of accuracy.
* The proposed method seems to be simple and generalizable.

Weakness:
* One weakness of this paper is the novelty of the proposed method. The method is to conduct magnitude pruning on activations for back-propagation.  This seems to be more like an engineering trick rather than a science / learning paper.
* This paper will be more solid if it’s validated on more downstream tasks.
* The theoretical results are a bit confusing: for example, in algorithm 1, it prunes activations during forward pass, yet in Lemma 1, it states that “so pruning the activation during backward does not change its gradient”.

---

> ### Author Response · Authors · 2022-08-02
> **Response to Reviewer agR8**
>
> ### 1. Novelty Concern
>
> We respectfully disagree this is an engineering trick. We think the engineering trick is describing the technique with very predictable performance. However, it is not clear whether the activations stored for backward can be sparse.  And our work has empirically and theoretically proven it.
>
> ### 2. More downstream tasks
>
> Nice suggestion. We want to refer you to the common response, where we include more experiment results on more downstream datasets in that section.
>
> ### 3. Explain the theoretical results
>
> Thanks for pointing this out. There is a typo for the sentence in Lemma 1, it should be “so pruning the activation stored for backward does not change its gradient”. We have addressed this typo in the updated version.
>
> ### 4. Typo in Figure 2
>
> Thanks for pointing this out. We have addressed this typo.
>
> ### 5. Memory difference between activations in forward pass and backward pass
>
> During the forward pass, the activations would be saved for the following backward pass. In contrast, during the backpropagation, it would only use the previous activation for computing gradient and the memory would not further increase. Therefore, the memory would reach the peak at the end of the forward pass during training.
>
> ### 6. How can pruning activations in backpropagation reduce memory by more than 2x
>
> To clarify, Back Razor happens in the forward pass. Specifically, it would prune the activation of layer $n-1$ and store the pruned version for backward when the forward proceeds to layer $n$.
> As discussed in the last question, the activations that require large memory usage are all in the forward pass, pruning in the forward pass can reduce memory by more than 2x.
>
> ### 7. Have you tried on language transformers?
>
> No, we haven’t. But this is a great suggestion, given the rebuttal time window is too short, we would extend Back Razor to language transformer tasks in the future version.

---

> > ### Author Response · Authors · 2022-08-05
> > **Look forward to further feedback**
> >
> > Dear Reviewer agR8,
> >
> > Since the discussion section has been started for a few days, it would be highly appreciated if you could look at the above responses and reply. In this way, if you still have concerns, we could have time to address them before due of the discussion section.
> >
> > We would also highly appreciate it if you could consider raising the score if our response has addressed your concerns.
> >
> > Thank you very much for your time and efforts.

---

### Author Response · Authors · 2022-08-02
**General Response to All Reviewers**

We thank all reviewers for their constructive comments. We updated the draft (the changes are marked in blue). Here, we add more experiments to support our claim as requested by reviewers agR8 and 3voy. And we also want to highlight these new results to all reviewers.

We start by extending the experiments on ProxylessNAS-Mobile with four more datasets. As shown in the following table, the proposed Back Razor can surpass TinyTL by [1.6, 4.8, 4.2, 2.4] for [Flowers, Cars, CUB, Food], respectively under comparable memory consumption. Moreover, it can also surpass TinyTL@320 for most datasets while saving more memory.
| Method                | Training Memory | Flowers  | Cars  | CUB  | Food  |
|-----------------------|-----------------|----------|-------|------|-------|
| FT-Last               | 31MB            | 90.1     | 50.9  | 73.3 | 68.7  |
| FT-Norm+Last          | 192MB           | 94.3     | 77.9  | 76.3 |  77.0 |
| FT-Full               | 366MB           | 96.8     | 91.2  | 81.3 | 83.8  |
| TinyTL                | 37MB            | 95.5     | 85.0  | 77.1 | 79.7  |
| TinyTL@320            | 65MB            | 96.8     | 88.8  | 81.0 | 82.9  |
| Back Razor@90% (ours) | 42MB            | 97.1     | 89.8  | 81.3 | 82.1  |

We also extend the experiments on ViT-B/16  with five more datasets. As shown in the following table, the proposed Back Razor can achieve comparable performance with FT-Full with much less memory usage. It is also worth noting that the proposed Back Razor can even improve the accuracy of CUB and Food. More baseline comparisons will be included in the future version.

| Method                | Training Memory | Flowers  | Cars  | Aircraft | CUB   | Food  |
|-----------------------|-----------------|----------|-------|----------|-------|-------|
| FT-Full               | 19235MB         | 99.5%    | 85.5% | 78.8%    | 85.5% | 90.3% |
| Back Razor@80% (ours) | 4565MB          | 99.4%    | 84.0% | 77.3%    | 86.3% | 90.5% |

---

### Meta-Review · Area_Chair_FVyn · 2022-09-05

**Recommendation:** Accept
**Confidence:** Certain

**Metareview:**

This paper focuses on pruning the backpropogation activation to reduce the memory footprint in transfer learning. The paper is well structured and the method is simple to understand. All the reviewers acknowledge that the experimental results are convincing. Overall, the meta-reviewer recommends acceptance of the paper.

**Award:**

No

---

### Decision · Program_Chairs · 2022-09-14

Accept